# Do Bio-Ethanol and Synthetic Ethanol Produced from Air-Captured CO_2_ Have the Same Degree of “Greenness” and Relevance to “Fossil C”?

**DOI:** 10.3390/molecules27072223

**Published:** 2022-03-29

**Authors:** Michele Aresta

**Affiliations:** Innovative Catalysis for Carbon Recycling, Via Camillo Rosalba 49, 70124 Bari, Italy; michele.aresta@ic2r.com

**Keywords:** carbon dioxide capture and utilization, bio-ethanol, e-fuels, renewable fuels of non-bio-origin

## Abstract

This paper discusses the epochal change in the reputation of carbon dioxide, which is now considered as a raw material alternative to fossil C for the synthesis of chemicals, materials and fuels, as opposed to a waste material that must be confined underground. In particular, its use as renewable C is compared to biomass. In this paper, a specific point is discussed: is ethanol (or any fuel) produced via the catalytic conversion of atmospheric CO_2_ different from the relevant biomass-sourced product(s)? The answer to this question is very important because it ultimately determines whether or not fuels derived from atmospheric CO_2_ (either e-fuels or solar fuels) have the right to be subsidized in the same way that biofuels are. Conclusions are drawn demonstrating that ethanol derived from atmospheric CO_2_ deserves the same benefits as bio-ethanol, with the additional advantage that its synthesis can be less pollutant than its production via the fermentation of sugars. The same concept can be applied to any fuel derived from atmospheric CO_2_.

## 1. Introduction

Nowadays, we notice a revolution concerning the reputation of the tiny CO_2_ molecule: from “*waste*” to “*resource*” [1]. I am one of those people who, for decades, has disseminated the concept that “*CO_2_ is a resource*” [2,3]. As a matter of fact, CO_2_ is at the origin of life, it was the original source of carbon for constructing organisms and life on our planet and still is the most abundant form of easily accessible carbon that we can dispose of (Table 1).

However, the atmosphere is the most abundant source of immediately available CO_2_, even if it is quite diluted (415 ppm). Subsurface deposits of CO_2_, less easily reachable but more concentrated (up to 99% pure), are usually close to volcanic areas and spread all over the planet. The amount of carbonate rocks is estimated to be *ca*. 10^9^
**Gt_C_** and varies over time because of the cycle of carbon (see Table 1 and Figure 1). The natural carbon cycle turns carbon from the atmosphere into plants, microorganisms or animals, and then returns to the atmosphere, soil or sub-surface (fossil C). Figure 1 shows the amount of carbon cycled yearly (*ca.* 200 **Gt_C_**/y).

## 2. The Need to Substitute Fossil C

The continuous extraction of fossil C and its combustion is causing an accumulation of CO_2_ in the atmosphere as the natural C cycle cannot buffer the excess 10 **Gt_C_** of anthropogenic origin, despite representing ca. 5% of the amount cycled yearly (Figure 1). Such an accumulation is considered to cause climate change. In my opinion, the increase in atmospheric CO_2_, which parallels the amount of burned C-based fuels by our society, is not the protagonist in the “*climate change drama*” played these days across the world in everyday life, but the third actor, with the inefficient use of chemical energy from fossil-fuels as the protagonist and the increase in atmospheric water vapor as the deuteragonist [1]. As a matter of fact, over 65% of the chemical energy of burned fossil C, due to the low efficiency of the conversion processes into electric, mechanical or thermal energy [7], is directly discharged into the atmosphere in the form of heat even at temperatures as high as 800–1000 °C, resulting in the direct heating of the atmosphere. This is causing an increase in atmospheric water vapor, which is a stronger greenhouse gas than CO_2_ [8]. The combination of all of these causes the adverse environmental events that we are currently observing, and may seriously affect our planet in the future. Such a negative impact could be mitigated by increasing the efficiency of energy production and a wiser use of energy. For example, IGCC technology has increased the *efficiency of the chemical-to-electric energy conversion* from an average of 32–35% to 55+%, causing a reasonable decrease of ca. 60% in the use of fossil C for electricity production, and thus CO_2_ emissions. Although very promising, such technology is rarely applied, despite its low costs [7].

Overall, individual and collective responsibilities play a role in improving the use of energy.

## 3. Alternatives to Fossil C

The alternative to the aforementioned efficiency technologies is a fossil C substitute that uses primary sources from non-fossil-C energy. The need to limit the use of fossil C as source of energy is now very urgent. After the almost unattended COP2016 Paris Agreement, hopefully, the urgency of an inversion of trend has been re-assessed by a large number of governments at the COP2021 in Glasgow. Moving away from fossil C also meets the goals of the 2030 Agenda for Sustainable Development that includes 17 Sustainable Development Goals (SDGs), first adopted by the UN General Assembly on September 2015. [8] As a matter of fact, we need to act in the direction of returning to nature, which does not produce waste and recycles carbon through the conversion of CO_2_. Although CO_2_ has been industrially used as raw material for over 150 years (aspirin and urea syntheses, the latter consuming more than 160 Mt_CO2_/y today) and CO_2_ chemistry has been intensively investigated during the last fifty years [1,9,10,11], attention is mainly devoted to the use of CO_2_ as a building block of chemicals, because its conversion into fuels requires energy and hydrogen, both of which are still mainly produced today from fossil C (>81% and 95%, respectively) [7,12]. Today, the large volume conversion of CO_2_ makes sense as it is possible without producing more CO_2_ than is used. As a matter of fact, the large-scale conversion of perennial energies (solar, wind, hydro, geothermal) into more easily usable forms of energy (electric and thermal) at affordable costs is making possible an energetic transition from fossil C to carbon-free energies for powering our lives [13].

In such an epochal transition, the utilization of CO_2_ as a building block of chemicals and/or source of carbon for energy products will play a key role. Carbon dioxide capture and utilization (CCU) is an active part of the circular economy strategy; at the same level, carbon dioxide capture and Storage (CCS) is the logical end-of-pipe technology of the linear economy. However, CCS and CCU are divergent technologies in the sense that, while the former pushes the extraction of fossil C, the latter avoids the extraction of fossil C. Therefore, CCS is somehow tied up with fossil-C-fed power plants, a moderately concentrated source of CO_2_ (5–14% *v/v* of flue gases), while CCU is attuned to direct air capture (DAC). CCS, which has received significant funding over the last thirty years, had to already be a large-scale technology for CO_2_ mitigation, but this was not achieved as several scientific gaps exist in its deployment (energetic costs for exploitation, permanence in disposal sites and environmental impact). However, in a world mainly powered by perennial energy sources, its role will be less and less important. As the energetic transition would not guarantee a future for energy produced by power plants, and because capturing CO_2_ from plants fired with fossil C would perpetuate the use of the latter, CCU correctly orientates towards DAC to make CO_2_ available for conversion, a technology that requires an energy and economic cost reduction (see below). On the other hand, CO_2_ is renewable C; biomass is made from it.

Moving to circularity represents a dramatic but necessary change in both the economy and our lives. The use of renewable carbon merges CCU and bioeconomy, with hybrid (chemo-enzymatic) catalytic systems boosting the potential of carbon recycling.

Three strategies can be envisaged for implementing such a major change in CCU: **i.** Large-scale non-fossil-H_2_ production and its use in chemo-catalytic CO_2_ reduction in fuels (e-fuels). **ii.** Coprocessing of CO_2_ and water under solar irradiation to afford energy products (solar fuels). **iii.** Integration of biotechnology and catalysis. In all cases, CO_2_ and water are at the core of the production of energy products; an economy based on CO_2_ and water [14]!

Strategy (i) is immediately deployed by using perennial energies (SWHG) to produce cheap and abundant electrons for the electrochemical reduction of non-drinkable water (recycled, salty) to afford hydrogen, which can then be used for the chemo-catalytic reduction of CO_2_ to gaseous (CH_4_) or liquid fuels (CH_3_OH, hydrocarbons, superior alcohols), using known technologies that operate in the Syngas-FT frame. This approach has some technological barriers that can be identified by the availability of: a. low-cost electrons, b. large-scale and long-life electrolyzers, and c. cheap and stable electrodes. It can be boosted by new technologies such as: d. the potential increase in the electrolysis temperature, e. working under pressure for immediate H_2_ distribution to users, and f. the exploitation of solid-state electrolysis. The use of renewable H_2_ for CO_2_ reduction into energy products may have a most immediate implementation, supposing that cheap sources of CO_2_ and cheap “electrons” (reducing power) are found to meet the cost target [15]. Fuels produced in this way, or fuels obtained from CO_2_ using non-fossil electricity and H_2_, are labeled “E-fuels”, the technology has a high TRL (7–9, depending on the targeted fuel: CH_4_ or CH_3_OH), and demo-plants are available in several countries. The bottleneck of this technology is its worldwide dissemination in the short term.

The second strategy (ii) is based on an advanced approach of co-processing CO_2_ and water to produce fuels without intermediate H_2_ production [16]. Such an approach has the great advantage of cutting the costs of producing, storing, transporting, and using H_2_, resulting in a much lower CAPEX, and even a lower OPEX, in terms of the safety measures to be implemented. A more probable approach is to use the Sun to power direct photochemical, photoelectrochemical, photobiochemical, and photobioelectrochemical routes of CO_2_ reduction and water oxidation [15], producing fuels that can be called solar fuels for simplicity [16]. The TRL of such technology is very low (2–3), but of great interest for the future. The third strategy (iii) is also at its early stage of development and combines electricity, enzymes, microorganisms and metal systems as catalysts for CO_2_ conversion into energy products and chemicals [17,18]. However, fossil fuels can potentially be substituted with biofuels, E-fuels, or solar fuels, each having their own peculiar character and potential to reduce carbon dioxide emissions. Because “CO_2_ avoidance” receives subventions from governments, unfortunately, such a large transition is increasing the discussions of how the new regulations and agreements can match their old counterparts or how older statements can be rephrased. Significantly, new non-fossil fuels enter into a confrontation with bio-sourced fuels or fuels derived from biomass, a class of products known for a long time and already on the market. It is worth recalling that biodiesel (produced from lipids, vegetal or animal sources) represents ca. 10% of the diesel used in several countries, while ethanol (produced via sugars fermentation, a very old practice) is added to gasoline in a 5–7% *v/v* ratio. All around the world, there are actions in favor of CCU: both calls for large-scale projects that support innovation actions at a high TRL (7–9) (Carbon Prize, USA–Canada; Green Deal and Innovation in the EU) and calls that address more basic research at a low TRL (EU Calls in Horizon) are funding the deployment of and search for innovative solutions that may make the knowledge we have for mimicking nature useful for building and deploying a *man-made C cycle*. Nowadays, at the EU level, there is a broad confrontation regarding the introduction of such new fuels into older schemes (transition from RED II to RED III), and understanding if they can be assimilated with biofuels and entitled to have similar benefits. This is not an easy task and requires agreement among different actors and lobbies (industrialists, farmers, fuel producers, policy men, etc.), because this task implements new rules and standards, avoiding the double counting of benefits as well as attributing benefits to unentitled goods.

## 4. Biofuels vs. Renewable Fuels of Non-Bio-Origin

Here enters the question that is the title of this paper: “Ethanol produced from biomass and ethanol produced catalytically from air-captured CO_2_: do they have a different degree of “Greenness” or “Fossil C””? The answer to the question is important. First of all, it is necessary to clarify if there exists any difference between *fossil CO_2_* and *bio CO_2_*, where *fossil CO_2_* is formed in the combustion of fossil C, and *bio CO_2_* is formed in the conversion (combustion/fermentation) of biomass. The use of biomass is seen as antithetic to the use of fossil C: the former avoids the extraction of fossil C, and thus avoids the production of new CO_2_ that will accumulate in the atmosphere. Using biomass (and its derived biofuels), is considered an action that does not overload the atmosphere with CO_2_ of fossil-fuel origin. Burning biomass or bio-fuels is considered a *zero-emission* option for energy production and use, even if it is not exactly so. In fact, accurate life cycle assessment (LCA) studies show that the use of biofuels in the current production–utilization–accounting scheme is the transfer of carbon from surface subsoil to the atmosphere, similar to the use of fossil C, even if it is much less intensive [19]. As a matter of fact, biomass is generated from atmospheric CO_2_, and when burned, it is believed to return to the atmosphere the same amount of CO_2_, as if the sequence was part of the natural C cycle. As a matter of fact, the cycle is not really closed as it occurs in nature. In fact, one should also consider, in addition to the carbon dioxide generated by the combustion of biomass, the amount of CO_2_ emitted by the various human activities that accompany the production and work-up of biomass (soil cultivation, planting, agricultural practices, the use of pesticides and herbicides, harvesting, converting the products into the final goods, producing biofuels, etc.), and even the soil carbon pauperization caused by agricultural practices. The latter aspect is often not considered, causing mistakes in assessment studies.

Nonetheless, biofuels are subsidized by governments (through tax breaks, grants, loans, and loan guarantees) for their environmentally friendly qualities and for the fact that they save natural fossil resources for future generations.

However, the question of whether converting anthropogenic CO_2_ into energy products is an action that mimics nature can be raised; therefore, such fuels can be considered as biofuels.

Let us start with clarifying what “*anthropogenic CO_2_*” is:

Anthropogenic CO_2_, aside from that emitted by humans during respiration (*ca.* 1 kg/d pax), is the amount of CO_2_ produced by burning biomass, biofuels or fossil C (such as coal, oil, gas) to produce energy or goods used by society, or even the part that accumulates in the atmosphere because of deforestation or is generated in forest fires. “*Anthropogenic CO_2_*”, therefore, encompasses all of the non-natural CO_2_ emitted into the atmosphere that overbalances the natural C cycle.

Let us now concentrate on two particular classes of fuels: (i) fuels derived from biomass (made from atmospheric CO_2_), and (ii) fuels made from CO_2_ emitted by industrial and chemical industries, cement manufacturing, stainless-steel manufacturing, or power plants. From these two cases, a very interesting and almost “philosophical” debate has been occurring for a long time. The difference between these two classes is not clear and requires a detailed and circumstantiated set of rules. 

On a general basis, both classes of fuels make fuels from CO_2_ and avoid fossil C, and their use can be considered beneficial. However, are they equal with respect to subsidies or subventions?

Industrial CO_2_ has its origin in fossil C that is fed to industries and fuels derived from it are, therefore, considered of “fossil” origin. Conversely, CO_2_ produced by burning biomass and its derivatives is said to be “biogenic”, and its emission does not cause, in principle, an increase in the atmospheric concentration of CO_2_ as the level of CO_2_ released should be fixed in the biomass. As we have discussed above, this is not completely true, and we must also consider that there is a gap of time between the combustion of bio-carbon and its fixation into biomass; its combustion is some 1000–10,000 times faster than biomass growth [1].

Moreover, the use of biomass-derived fuels produces less-intense CO_2_ emissions than the combustion of fossil C. However, industrial CO_2_ (fossil C-derived) has different characteristics compared to biogenic CO_2_, the latter being produced by actions of microorganisms in biomass or in the process of burning bio C.

Let us go further and conduct a more detailed analysis of the relationship between the origin of CO_2_ and its conversion into fuels. 

At the end of the 1990s, at the EU level, during the very early days of the question of CO_2_ recovery and conversion (the “Recovery and Utilization of Carbon Dioxide-RUCADI” project was the first EU-funded project for CCUS, founded in 1998 and co-ordinated by M. Aresta), and at the beginning of the discussion about the classification of fuels and goods derived from CO_2_, a scenario was depicted for the consideration of scientists, lawyers and policy makers. 

*Scenario*: Let us suppose that we have a source of CO_2_ from fossil fuels (power plants) and the stream follows two separate routes. *Route 1*: CO_2_ from the power plant is emitted into the atmosphere and then a pond of algae, located somewhere, fixes the aerobic CO_2_. *Route 2*: the same CO_2_ flow is, instead, the only C source, and is fed directly into a pond of algae located next to the power plant. Let us now extract lipids from both batches of algae and convert them into diesel by using the same technologies, obtaining *diesel1* and *diesel2*.

*Question*: Are the *diesel1* and *diesel2* produced in the two ponds different? Are they distinguishable? Could both be labeled “biodiesel”? Do the two fuels have the same rights to be subsidized?

This may appear a dull question: two different opinions commonly arise: (i) “Biomass is biomass, however it is grown”, so the two diesels are equal. (ii) If we feed fossil CO_2_, we produce fossil fuels, so the two diesels are different.

Then, the question arises: if they are different, how can we distinguish them? This question is easy to answer: the ^14^C analysis will clearly show which is which: fossil CO_2_ and atmospheric CO_2_ have a different ^14^C level. *Diesel1* and *diesel2* are, therefore, different.

The most recent conceptualization, based on a broad consultation and averaging different points of view, arrives at the same conclusions but moves from a quite different position: using CO_2_ captured from a fossil C-fed power plant (or industrial plant) supports the continued extensive use of fossil C, and this cannot be subsidized. CO_2_ captured from industrial and power-generation emissions is totally different from bio CO_2_. Only bio CO_2_ can be subsidized, and the products derived from it can be considered of bio-origin and called bio-fuels.

Now we present a different case. CO_2_ produced by burning fossil C is not captured but emitted into the atmosphere. Then, ethanol is made, or any other fuel, following two routes. *Route A*: Corn is grown, and then ethanol is produced by fermentation. *Route B*: CO_2_ is taken from the atmosphere, and ethanol is produced catalytically in a chemical plant. Are *ethanolA* and *ethanolB* different? The technique mentioned above (^14^C isotope abundance) will not distinguish the two: in both cases, atmospheric CO_2_ is used. One can say that the two butches of ethanol will still have different analytical characters: (A) will most likely contain proteins and other biocomponents that will reveal its bio-origin; (B) will not have such properties. However, such differences are due to the different production technologies and not the different origins of CO_2_. Moreover, ethanol B is cleaner than ethanol A.

*Then, is the corn-sourced ethanol different from synthetic ethanol when atmospheric CO_2_ is used for the production of both?* Is this a new sophism? If one looks at the source of CO_2_ (in both cases, this is the atmosphere), they are not different. It is not the fact that one is produced using a vegetal plant and the other using a chemical plant that can differentiate between them. The common source of carbon (atmospheric C) makes them equal: both have the same link to fossil C, and both recycle atmospheric carbon. Intriguingly, their chemical synthesis is not seasonal, and the rate of ethanol production and the volume produced per unit area of the occupied land would be higher than for the bio-process. The chemical synthesis would be more intensive and present other advantages over the biological route, as arable soil would not be required, C soil pauperization would not occur and much less water would be required per unit volume of ethanol. *Developing synthetic procedures based on the utilization of atmospheric CO_2_ (man-made C cycle) would be a win–win situation* that may side with the *natural C cycle* and alleviate the impact of the use of fossil C by progressively reducing its demand.

The tendency is to capture CO_2_ from the atmosphere as the only source that guarantees infinitely large volumes of immediately available CO_2_, as stated above: fuels derived from such CO_2_ should not be distinguished from bio-fuels; they should be considered as “*renewable fuels of non-biological origin, RF-NBO*” and subsidized in the same way that biofuels are.

In a world in which primary energy will mainly be provided by perennial sources (SWHG), synthetic fuels obtained from atmospheric CO_2_ may be more environmentally beneficial than biofuels. Using recyclable heterogeneous catalysts with airborn CO_2_ and non-drinkable water as hydrogen sources will produce less pollution than growing biomass (this requires high-quality soil, nutrients for soil with the emission of N compounds and other pollutants, in addition to agrochemicals, such as herbicides and pesticides, and large volumes of water) and a conversion into bio-fuels by using biotechnologies based on fermentation by microorganisms.

Now, let us tell this story with numbers. To make 1 L of ethanol (789 g, 17 mol) from corn, some 10–17 L [20] of clean water is necessary (equal to 10,000–17,000 g; or 554–940 mol) solely for farming and not considering the water needed for fermentation and work-up. Each mol of ethanol is will, thus, requires 32.6–55.4 mol of water. To make ethanol from air-captured CO_2_, according to Equation (1), assuming an efficiency in water electrolysis to H_2_ of 80% and a chemical yield of 60%, one needs only 12.5 moles of water per mol of ethanol produced, saving from 20 to 43 mol of water per mol ethanol, or 7.8–12.9 t_H2O_/t_ethanol_: a huge amount considering that today the consumption of bioethanol is close to 50 Gt/y. Moreover, the watering of soil requires soft water, while hydrogen can be produced from salty water, a large difference that will save on large volumes of high-quality water:2 CO_2_ + 6 H_2_ → CH_3_CH_2_OH + 3 H_2_O(1)

Considering the energy necessary for producing bio-ethanol and synthetic ethanol, the latter made from DAC-CO_2_ and H_2_ from PV–water electrolysis, one finds that the non-optimized energy for bioethanol production is 1.316 MJ/mol [21], not accounting for soil C pauperization, while synthetic ethanol requires 2.50 MJ/mol [22,23]. The former value might be optimized to 0.92 MJ/mol by using the most well-known technologies [21], not accounting for soil restoration, which remains an ongoing debate, while the latter can be more than halved to 1.00 MJ/mol by: *i.* increasing PV efficiency (solar to electrons) from 20% to a perspective of 40% by 2040 [1]; *ii.* reducing the cost of PV electrons by using cheaper materials for PV (organic materials instead of Si-based materials); *iii.* improving the electrolyzer size, life and cost; *iv.* implementing heat recycling; and *v.* improving the DAC technology, that today is very expensive, both economically (ca. 200 EUR/t_CO2_) and energetically (0.06–0.16 MJ/mol_CO2_) [24]. Therefore, one can say that, by 2040, the energy necessary for making bio- or synthetic ethanol will be almost the same, with the great advantage that synthetic ethanol can be produced everywhere without climate constraints and will require much less water of a lower quantity, a great positive. Another advantage of the synthetic route is that will require less space for producing the same amount of ethanol with respect to the bio-route, and lower quality soil. A drawback of bio-ethanol is that it will produce large volumes of waste biomass that must be used (production of thermal energy, or further working up to produce sugars to increase the amount of ethanol) for not aggravating the energy balance. In the synthetic route, heat recovery will improve the overall energy balance, and developing selective catalysts will avoid the production of side-products and loss of carbon. The industry will require a higher CAPEX with respect to growing corn, that can be significantly reduced by retrofitting and revamping existing plants.

Notably, both CO_2_ and water can be recovered from the atmosphere with the benefit of reducing the concentration of two greenhouse gases, and this will make the synthesis of ethanol ubiquitous and not linked to a specification of soil and climate. 

As neither route (bio or synthetic) prevails in a net manner and both have *pros* and *cons* (Table 2), considering that DAC-CO_2_ cuts out debates on the origin of CO_2_ and that synthetic ethanol is commonly labeled as a renewable fuel of non-bio-origin (RF-NBO), both routes can be equally useful for producing a liquid fuel that may have widespread applications in the land transport sector, or it could be used as raw material in the fuel industry for producing longer-chain hydrocarbons [25], possibly reaching the state of fuel usable in the aviation sector.

## 5. Conclusions

A revolution is in front of us, based on an industrial and energetic transition that will progressively reduce the use of fossil C most likely to 30% of the current use by 2040–2050 (in my view, fossil C-based fuels will still be used till the end of this century, even if at a much lower rate), and technologies closer to nature will be implemented, improving the quality of our lives and preserving our planet. Table 3 lists the potential substitutes for fossil C and its actual production/or TRL.

As biomass cannot satisfy the energy needs of our society (this has been the case since the start of the Industrial Revolution), developing man-made technologies that can complement the bio-production of chemicals and energy products is of fundamental importance. The use of atmospheric C in both cases will greatly reduce the environmental impact of anthropic activities and mitigate climate change.

A key point that the regulations have to make clear is that subventions and benefits cannot be given twice to the same or to different recipients for the same action: either the carbon capture is subsidized or captured CO_2_ is converted into marketable products.

## Figures and Tables

**Figure 1 molecules-27-02223-f001:**
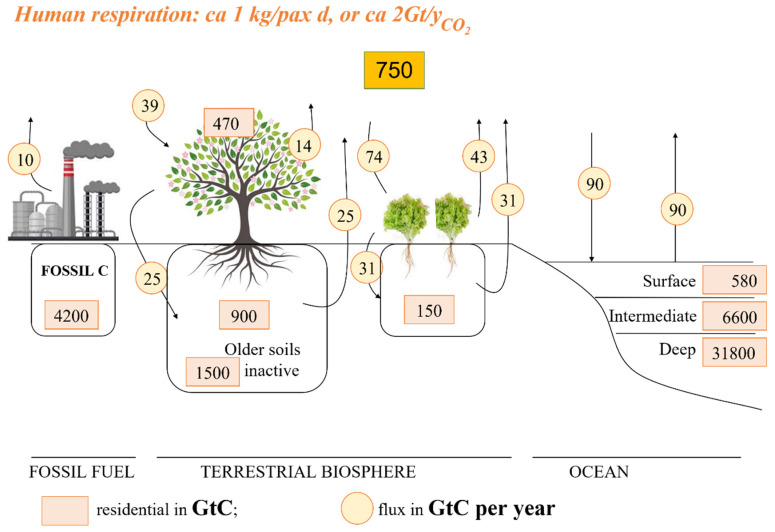
The natural C cycle.

**Table 1 molecules-27-02223-t001:** Distribution of carbon among different environments on Earth (**as Gt_C_**).

Atmospheric CO_2_	750
Biosphere [4]	550–560
*Plants*	*450–460*
*Animals*	*2 (Humans 0.06)*
*Bacteria*	*70*
*Archaea*	*7*
*Fungi*	*12*
*Protists*	*4*
*Viruses*	*0.2*
Carbonate rocks ^a^ (sediments) [5]	1.8 × 10^9^
Fossil carbon [1]	
*Coal*	*607*
*Oil*	*167*
*Natural Gas*	*167*
CO_2_ deposits (degas rate 0.1 **Gt_C_**/y)	30,000
Ocean deep floor [6]	37,000

^a^ During the Phanerozoic aeon (system of rocks deposited during the Phanerozoic era—the Paleozoic, Mesozoic and Cenozoic eras—from 541 My to present), 2100 × 10^15^ t of carbonate rocks were deposited with a mass cycle of 8.6 × 10^14^ t_CaCO3_/My and a decay constant of 0.0025 My^−1^; 43,500 **Gt_C_** above surface equal to 10–15% of all rocks.

**Table 2 molecules-27-02223-t002:** Pros and cons of the *bio* and *synthetic route* to ethanol.

Category	Bio-Ethanol		Synthetic	
	Pros	Cons	Pros	Cons
Soil fertility		High	Low ^1^	
Soil C-use		High ^2^	NO	
Land extension		High	Low	
CAPEX	Low			High
OPEX		High	Low	
Energy consumption	Neutral ^3^		Neutral ^3^	
GHG emission		Agrochemicals ^4^		
Water consumption		High	Low	
Geographical location		Climate dependence	Any place	
Production cost	Neutral		Neutral	
Waste production, recovery and utilization		Waste biomass utilization		Heat recovery

^1^ Marginal areas can be used; ^2^ Agricultural practices cause soil C pauperization; ^3^ In the future, both practices may have levelled energy requirements; ^4^ The production/use of agrochemical cause large emission of GHGs.

**Table 3 molecules-27-02223-t003:** Potential of biofuels and E-fuels as substitutes of fossil C-derived fuels.

Fuel	World Volume Consumed/y or *TRL*	Expected Consumption by 2030	Ref.
Bio-ethanol	132 BL in 2020	137 BL	[26]
Bio-diesel	48 BL in 2020	50 BL	[27]
Bio-jet fuel	15 ML in 2020	500 ML	[28]
Bio-Gas	31 Mtoe in 2018	41 Mtoe; 78 Mtoe	[29]
Bio-methane	1 Mtoe in 2018	46 Mtoe; 114 Mtoe	[29]
e-H_2_	320 kt	73–158 Mt	[30]
e-CH_4_	*7–9*	Depends on e-H_2_ cost	[31]
e-CH_3_OH	*600 t*	Depends on e-H_2_ cost	[31]
S-fuels	*TRL 2–3*	Research is needed	[19]

(BL = billion liters; ML = million liters; Mtoe: million ton oil equivalent; Mt = million tons; TRL: Technological readiness level, 1–9 scale).

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
