# Peer review of "Do Bio-Ethanol and Synthetic Ethanol Produced from Air-Captured CO2 Have the Same Degree of “Greenness” and Relevance to “Fossil C”?"

_molecules, 2022, doi:10.3390/molecules27072223_

Round 1

Reviewer 1 Report

The scientific area of this paper is very important at the present time. The main goal was defined precisely, and the manuscript is written very well. 
I suggest only a some very minor revisions such as:
+ please correct the arrow at scheme (1)
+ please to improved the quality of the figure 1

Author Response

Thanks for the positive comments. Figure 1 has been changed and Scheme 1 amended.

Reviewer 2 Report

I do agree that a revolution is in front of us, based on an industrial and energetic transition that will progressively reduce the use of fossil-C.

Mostly, the paper is well written, deals with one of the most important topics.

I cannot find any drastic mistakes,  thus suggesting accepting the paper in its present form.

Author Response

Thanks for the very positive evaluation of this paper

Reviewer 3 Report

This  paper presents on the epochal change in the reputation of carbon dioxide, which could be a substitute for the conventional fossil-carbon for the production of chemicals, and fuels. This is an interesting piece of work however, there are several comments required to be addressed by the authors before it could be considered to be accepted for publication.

The comments are as follows:

  1. Keyword needs to be more specific. Please revise it.
  2. Figure 1 is a bit too messy and crowded with information. You are required to do necessary adjustment to it.
  3. Section 3 - It would be great to include some initiatives to address the UN SDG and sustainability aspects in this section.
  4. Section 4 - Please include a table on the summary of recent literatures and key findings for the renewable biofuels.

Author Response

Thanks for your positive evaluation of the paper. The following changes have ben implemented.

i. More keywords added

ii. Figure 1 has been changed

iii. UN SDGs have been mentioned in the text.

iv. Table 3 has been added with the actual and perspective use of renewable fuels